# Secondary metabolites of the invasive tree, Callery pear (*Pyrus calleryana*), provide support for the empty niche theory of invasion

Jessica A. Hartshorn¤*

Forestry and Environmental Conservation, Clemson University, Clemson, South Carolina, United States of America

¤ Current affiliation: Agriculture Research Development Program, Central State University, Wilberforce, Ohio, United States of America
* jhartshorn@centralstate.edu

## Abstract

Invasive woody species like Callery pear (*Pyrus calleryana* Decne.) alter ecosystems directly and indirectly through effects on arthropod communities as well as chemical alterations of the soil. Evidence suggests that the aggressive spread and negative impacts are due to allelopathic chemicals present throughout plant tissues which reduce herbivory and add unique allochthonous inputs to the soil, thereby reducing germination of native species and furthering Callery pear's domination on the landscape. To assess the allelopathic potential of Callery pear, we collected leaf tissue from Callery pear, black cherry (*Prunus serotina*) as the native comparison, and wild peach (*Prunus persica*) to serve as a non-native but non-invasive representative. Callery pear leaves contained 32 unique compounds compared to our other two species tested, with 23 of these compounds belonging to the group of compounds called flavones, which are secondary metabolites known to build up in the roots of plants and cause autotoxicity, along with changes to the soil microbial community, including mycorrhizal fungi. While flavones are ubiquitous in nature, their presence in Callery pear suggests downstream effects on native plants and arthropod communities, and provides evidence for the empty niche theory of invasion. Further experiments are needed to confirm flavones in other Callery pear tissues and to assess the mediating chemical pathways that lead to their upregulation.

## Introduction

Invasive woody species cause billions of dollars annually in damage to economically important commodities, and untold damage to ecologically important species and habitats [1]. Invasive plants can affect ecosystems through bottom-up processes such as effects on herbivores and higher trophic level communities [2] and through top-down processes such as changes in soil chemistry due to allelopathic chemicals

provided the original author and source are credited.

**Data availability statement:** All metabolomics master data files are available from the github database (https://github.com/jhartsh/Callery-pear-data.git).

**Funding:** The author(s) received no specific funding for this work.

[3]. Allelopathy in the soil comes from the production of secondary metabolites which build up in root tissues which are then sloughed off into the surrounding rhizosphere [4]. These alterations to the soil due to invasive woody plants like common privet (*Ligustrum sinense*) have been shown to negatively impact soil-dwelling organisms such as native earthworms, and its removal leads to improvement in native arthropod and earthworm communities [5,6].

Secondary metabolites are ubiquitous in that they are produced by all plants, but are also incredibly varied, with estimates of 10,000 unique varieties across plants [7]. They also tend to give plants their unique signatures of odor, taste, and color, and are commonly used in products meant for human consumption. Secondary metabolites can be produced through multiple pathways, and some secondary metabolites like flavonoids may be produced through different biochemical pathways in the same plant at the same time [8]. These chemicals may be present only when under attack or may be upregulated under stressful conditions, such as abiotic disturbance or when challenged by a pest or pathogen [8]. Many hypotheses have been proposed to explain the success of non-native invasive plants; namely, the Novel Weapons Hypothesis [NWH; [9]] has been used to explain invasions through the production of these secondary metabolites which native plants are unable to metabolize. In support of this hypothesis, non-native invasive plants have been shown to have more unique secondary metabolites compared to native congeners [10].

Callery pear (CP; *Pyrus calleryana* Decne.) is an invasive tree that was originally introduced to the United States in attempts to alleviate fireblight (*Erwinia amylovora*), a bacterial disease that was decimating the European pear (EP) industry [11]. Despite the lack of success in cultivating a commercial pear variety from CP rootstock, many other varieties were created and planted as ornamentals throughout residential areas in the eastern U.S. with the 'Bradford' variety being the most well-known [12]. Since the 1990s, CP has become dominant across much of the Southeastern landscape, with infestations spanning the eastern half of the U.S. [12]. Due to a lack of natural enemies found feeding on Callery pear, hypotheses such as the Enemy Release Hypothesis (ERH) [13] have been proposed to explain CP's exponential spread throughout the U.S. Additional research suggests that the aggressive spread of CP is due to its negative effects on germinating natives, with allelopathic chemicals being the suggested mechanism [14]. In response to research demonstrating these impacts, several states such as Ohio and South Carolina have banned the importation and sale of *P. calleryana* [15,16] with programs like the 'Bradford Pear Bounty' serving to remove standing CP from the landscape [17].

Phylogenetic relationships have been used to predict invasiveness of plant species, under the assumption that functional traits leading to successful invasion are closely tied to taxa closeness [18] but this does not delve into the potential underlying mechanisms allowing some species to dominate and not others. Two contradictory theories exist to explain the role of phylogenetic closeness in invasions; neutral theory [19] suggests that functional traits and phylogeny are independent and the spread and impacts we associate with invasive species are due to stochastic processes. Conversely, empty niche theory [20] is a concept borrowed from community

ecology [21] that suggests that phylogeny is inextricably connected to a plant's function in the environment, and so non-native species which are phylogenetically close to natives are at a disadvantage due to competitive exclusion by native congeners.

While CP has garnered much attention for its negative environmental impacts, other non-native woody Rosaceous trees such as peach (*Prunus persica* L. (Batsch)) have been grown and sold across many states in the U.S. since the 1700s, is the second most widely cultivated temperate-zone tree fruit, second only to apple (*Malus*), yet neither is considered invasive in any of the United States [22,23]. Conversely, the European pear (EP; *Py. communis* L.) has been cultivated for about the same length of time as peach, with virtually all cultivation occurring in three states (Washington, Oregon, and California), and has been reported as escaped in many states and has been listed as invasive in the state of Oregon [24]. While both *Prunus* and *Malus* genera have North American congeners, there are no native *Pyrus* in the Western Hemisphere, suggesting evidence for niche theory of invasion.

Understanding the drivers behind successful invasion of woody species is an important mediator in determining proper management and mitigation of current invasives, as well as predicting, detecting, and monitoring new or suspected invasives. Currently, the driving mechanisms behind the spread and invasion of CP are unknown and, as such, it is expected to continue causing ecological and economic damage. Identifying these mechanisms will also provide evidence to support, or not, long-standing theories of invasion. To address this gap in knowledge, I compared the secondary metabolite profile of CP, peach, and black cherry and predicted that the profile of CP would be significantly different from that of both peach and black cherry.

## Methods

### Sample collection

On 13 June 2023 I collected mature leaf tissues from three species of trees [CP (n = 14), wild peach (n = 11), and BC (*Prunus serotina;* n = 15)] from eight sites across Pickens and Oconee Counties in South Carolina (Table 1, Fig. 1).

**Table 1. Sample collection sites.**

| Site name Lat Long | Callery pear (Pyrus calleryana) | Black cherry (Prunus serotina) | Wild peach (Prunus persica) |
|---|---|---|---|
| Radio Station (34.682442, −82.988414) | 0 | 1 | 7 |
| Martin Creek (34.629216, −82.902454) | 1 | 2 | 1 |
| Seneca Creek (34.661928, −82.871712) | 0 | 2 | 1 |
| Pier (34.652219, −82.864539) | 8 | 0 | 0 |
| Ag Services (34.660532, −82.830898) | 0 | 1 | 0 |
| Rock Creek (34.675489, −82.808936) | 0 | 1 | 2 |
| Ingles (34.702969, −82.794050) | 5 | 3 | 0 |
| Disc golf (34.711556, −82.783614) | 0 | 5 | 0 |

Site names along with coordinates and the number of each species sampled at each site.

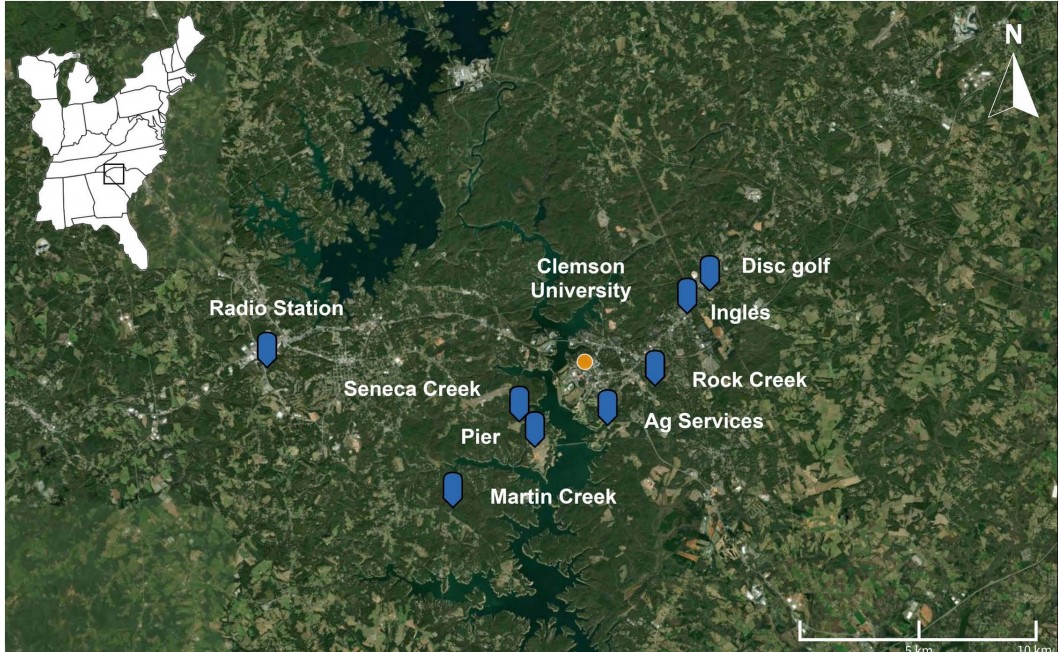

**Fig 1. Map of sites along with site names for all samples collected during the study.**

Pickens and Oconee Counties are classified as humid subtropical climate and receives approx. 143.9 cm (56.6 in.) of precipitation each year [25]. Soils in this area fall under the Starr series which consists of very well-drained and deep loamy soils with slopes ranging from 0–8% [26]. All sites were in periurban or rural areas and were adjacent to forested land which mainly contained loblolly pines (*Pinus taeda* L.) along with mixed hardwoods (e.g., *Carya* and *Quercus*). Nearby forested lands are also used regularly for recreation and contain mountain biking trails, disc golf courses, and other public use features. Because of the nature of these sites as public lands, and collections being limited to punches from leaves, no permits were required for sample collections.

We collected a minimum of 10 samples per tree from 4–5 leaves scattered across the outer edges of the canopy using a hole punch that was sterilized between samples with 70% EtOH. Punches were immediately placed into a pre-labeled microcentrifuge tube which was then placed inside a styrofoam container with dry ice to flash freeze the tissue. All tubes were then stored in a -80C freezer on Clemson campus until processing.

## Sample preparation

Frozen hole punch samples were weighed and then extracted using 80% methanol at a ratio of 1:10 mg/μL (tissue:solvent) which was chosen based on preliminary extractions and instrument analyses of a subset of samples focusing on highest number of mass features, while avoiding detector oversaturation. After extraction, samples were homogenized by adding 15–20 ceramic beads per sample and centrifuging them at 7,200 rpm for 15 cycles (30 sec./cycle at 4C) and then a final cycle of 12,000 rpm for 5 mins. at 4C. Resulting supernatant was then transferred to individual 2mL Eppendorf tubes which were stored at -80C until metabolite profiling. To maintain quality control (QC), a 45 μL aliquots of the resulting supernatants from each sample was mixed with 5 μL of the internal standard mix (10 ug/mL resveratrol-13C6 and 40x diluted reference material) in a 2 mL tube and labeled as the "Pooled QC". The final internal standard concentration was 1 μg/mL of resveratrol-13C6.

## Metabolite profiling

Identification of compounds was performed with UHP liquid chromatography using an Orbitrap Fusion Tribrid Mass Spectrometer (Thermo Scientific, Waltham, Mass., USA) LC-MS/MS with a sample injection volume of 2 µL. Data was then processed using Compound Discoverer (Version 3.2) for non-targeted metabolomics analysis. Compounds were annotated with online and in-house accurate mass libraries (ChemSpider, etc.) and MSn mass spectral libraries (mzCloud, mzVault with in-house and public databases). Level 1 compound annotations were performed with analysis of authentic reference standards using the same instrument method to obtain accurate mass, MS2 fragmentation, and retention time matches (S1 Table). Mass features detected were filtered using the data filters below to remove false signals and artifacts of the instrument analysis.

In LC-MS/MS analyses, some compounds produce multiple mass features during the ionization process including in-source fragments (ISF) ([M-Hexose+H]+, [M-H2O+H]+,etc.), adduct ions ([M + Na]+, [M + NH4]+, [M + K]+, etc.), and multimer ions (dimers, trimers, etc.). High quality data analysis involves annotation of these alternative features and this was done using online and in house MS2/MS3 spectral libraries (mzCloud or mzVault), accurate mass and structural databases, and *in silico* fragmentation predictions with FISh scoring in Compound Discoverer. Reported confidence levels (1–5) for annotations is based on the classification schema described in Schymanski et al. [27].

## Statistical analyses

Because this study involved non-targeted analysis, absolute quantification of compounds is not possible. To assess the different compounds between species, a partial least squares discriminant analysis (PLS-DA) was used to calculate VIP scores which were then used to identify compounds that were significantly different between CP and native BC, as well as between CP and non-native wild peach. After performing an initial PLS-DA, one sample of CP was identified as an outlier and removed from the analysis before conducting a second PLS-DA. All statistical analyses were performed with MetaboAnalyst online software (version 5.0) [28].

## Results

A total of 2,352 total compounds were identified across all three species of trees collected (n = 40), of which 1,861 (79%) were classified as level 4, 92 (4%) were classified as level 3, 131 (5.6%) were classified as level 2, and 180 (7.6%) were ISF adducts. After removing all non-level 1 compounds, this resulted in 78 compounds (3%) which were significantly different between Callery pear and peach, and Callery pear and cherry, but not peach and cherry. After removing redundancies this left us with 32 total compounds, 23 (72%) of which are flavonoids (Table 2).

The PLS-DA (Fig. 2) shows a clear distinction between the three species collected in this study. VIP scores of *Pr. persica* and *Pr. serotina* are more closely related to one another with respect to component 1, while *Py. calleryana* and *Pr. serotina* are more closely related with respect to component 2. However, component 1 accounts for a much higher amount of variation (51.5%) compared to component 2 (28.8%).

## Discussion

This study was the first to identify the secondary metabolite profile of CP and compare the identified compounds with a native species (black cherry) and a non-native, but not invasive, species (wild peach). There was a total of 32 compounds which were significantly different between CP, wild peach, and black cherry, with 23 of these belonging to the class of phytochemicals called flavonoids. The compounds identified as significant in CP tissue are the same as those identified from leaf tissue of the EP [29], however, as previously mentioned, it is not possible to compare absolute concentrations due to the non-targeted approach of this study. However, the invasiveness of EP along with the invasiveness of CP, when grown and distributed in different parts of the U.S. does suggest that these secondary metabolite compounds affect their

**Table 2. Unique secondary metabolite profile of Callery pear (*Pyrus calleryana*).**

| Compound | Chemical Class | Notes & Chemical Properties |
|---|---|---|
| Apigenin | Flavone | Aglycone of glycoside; Antioxidant, anti-inflammatory |
| Catechin | Flavonol | Antioxidant |
| Cosmosiin (Apigetrin) | Flavone | Antioxidant |
| Diosmetin | O-methylated flavone | Commonly found in citrus fruits; Antioxidant and antimicrobial |
| Hyperoside | Flavonol glycoside | Anti-inflammatory and antimicrobial |
| Isorhoifolin | Flavonoid glycoside | Aglycone form of apigenin; antioxidant |
| Luteolin | Flavone | Antioxidant and anti-inflammatory |
| Manghaslin | Flavonoid (glycoside) | Anti-inflammatory |
| Rosmarinic acid | Polyphenol | Antioxidant |
| Procyanidin B2 | Hydroxyflavan | Antioxidant and anti-inflammatory |
| Luteolin-7-O-glucoside | Glycosyloxyflavone | Glycosylated form of Luteolin; Antioxidant |
| Luteolin | Flavone | Antioxidant |

List of compounds which were identified as unique or significantly higher in Callery pear (*Pyrus calleryana*) compared to black cherry (*Prunus serotina*) and wild peach (*Pr. persica*) along with the respective chemical class and brief description of chemical properties.

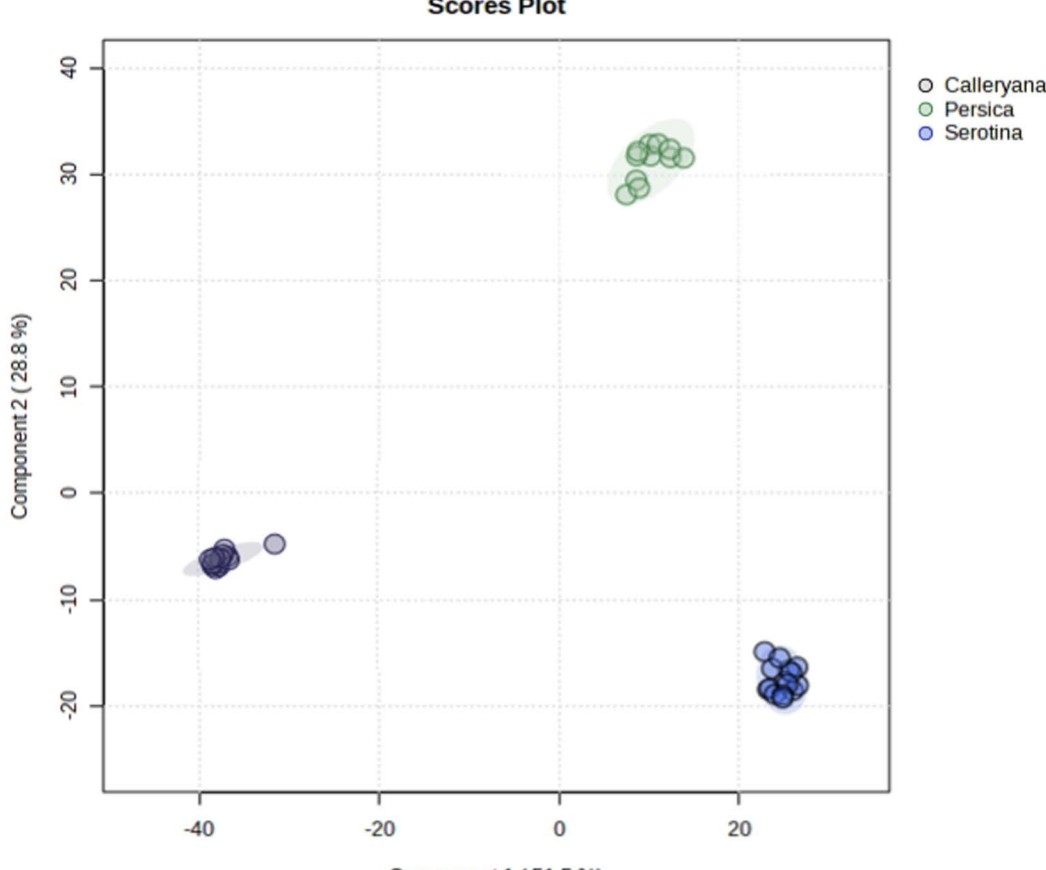

**Fig 2. Partial least squares discriminant analysis (PLS-DA) of compound VIP scores from field-collected samples.** Circles indicate individual samples with colors indicating species (*Py. calleryana* = grey, *Pr. persica* = green, *Pr. serotina* = blue).

ecological function and, therefore, their invasiveness in North America. These results provide support for the empty niche theory of invasion, whereby the compounds produced are related to their function in the environment and that the lack of native *Pyrus* species in the U.S. provides an empty niche for these non-native species to fill. Next steps to further investigate the role of these secondary metabolites are to target analyses to these 32 compounds for quantification in CP plus EP along with natives like black cherry and non-native non-invasives like wild peach.

Flavonoids are ubiquitous throughout nature and are found throughout all plant tissue types from reproductive structures to stems to roots [7]. Flavonoids are produced via the phenylpropanoid or acetate-malonate metabolic pathways, with the latter being far less studied, and have precursors which are chemically similar to other compounds used during the process of lignin production [4]. The multiple pathways of flavonoid production result in a variety of different structures including flavanones, isoflavonones, isoflavans, and pterocarpans [30].

Flavonoids are incredibly important compounds which serve as antioxidants to protect plants from herbivory, UV damage, and phytopathogens, and even affect the availability of certain nutrients in the soil, especially metals like iron [8]. They are also involved in the production of fatty acids and, therefore, can function as both a primary and a secondary metabolite, although little has been done to investigate the role of flavonoids in the primary metabolism of plants [31].

Flavonoids are especially important in roots, where they accumulate in the cells of the root cap and are eventually sloughed off into the soil to act as allelopathic chemicals [4]. Very few studies have quantified the movement of flavonoids into the rhizosphere from root cells of surrounding plants, however, we do know that exudation of flavonoids can increase in response to signaling of microbial symbionts as well as pathogens, and even abiotic stress [8]. This signaling is typically to benefit the allelopathy producer; flavonoids can attract Nitrogen-fixing bacteria in the soil and promote the establishment and growth of mycorrhizal fungi [32]. Flavonoids can also serve to inhibit organisms which may attack the plant in question. For example, kaempferol, a compound found to be significant in this study, has been shown to inhibit the motility of the root knot nematode, *Meloidogyne incognita*, thereby also affecting hatch rates and migration patterns [33].

Flavonoids can also inhibit soil-borne root pathogens as well, especially fungi. Specifically, isoflavonoids, flavans, or flavanones are potent anti-microbials and act as toxins to microbial pathogens through inhibition of the fungal germ tube and elongation of mycilial hyphae [4]. During attacks by these pathogens, it's thought that the antimicrobial compounds then become oxidized which then form free radicals, stimulating cell death during the host's hypersensitive response [4]. These compounds are also rapidly released during decomposition, and autotoxicity of the producing plants has been found, with the potential explanation being to prevent competition between mature plants and young seedlings by inhibiting sprouting. This is the opposite of what is seen with CP, which regularly produces stump sprouts and will even produce new shoots from stems that have been bulldozed and uprooted. Future research should directly identify the effects of CP-produced flavonoids on native plants, as well as non-native but not invasive plants, by applying these flavonoids directly to plants to quantify the effects on growth and defenses. Additionally, it is necessary to compare the rhizosphere microbial community in CP to other species to quantify changes in decomposers in the soil.

There are several limitations to consider when interpreting these results. First, samples were collected in midsummer and are not representative of the full range of seasonal changes to phytochemical production. Future studies should aim to collect tissue from early, mid, and late growing season which may show changes to the secondary metabolite profile. While trees were selected from similar sites which had a history of disturbance and were located in harsh environments, it is necessary to collect a wider range of samples from sites with different disturbance regimes and intensities. Along these same lines, future studies may find different results when collecting tissue samples from CP growing as a monoculture in an abandoned field versus CP growing in the understory of a mature pine plantation. Owing to CP's wide range of habitats, soil conditions, and growth patterns, I was not able to sample from the full range of potential growing sites and this is an area that requires further investigation. Finally, leaf tissue is important when it comes to leaf litter accumulation and the potential for secondary metabolite leaching into the soil, however, future studies should collect tissue from different parts of the tree to compare secondary metabolite profiles from various tissue types as well.

## Conclusions

Callery pear is a destructive woody plant which causes significant economic and untold ecological damage where it has invaded. Its unique place as an invasive Rosaceous tree has wide reaching implications for commercial industries as well as conservation of natural areas. A better understanding of the drivers behind its continued spread and impacts is necessary, not only to manage current infestations, but to prevent and predict future ones as well. Through non-targeted secondary metabolite analysis, I found that CP produces the same compounds as other invasive Rosaceous trees and propose that these results support the niche theory of invasion. However, other biological factors of CP invasion such as release from enemy pressure also likely contribute to its successful invasion. Further research is needed to compare compounds over the growing season, in different habitats and tissue types, and in the presence of native species which are being negatively affected by CP. With this additional information, researchers can begin piecing together the different components of invasion and associated management options to mitigate further spread.

## Supporting information

**S1 Table. Identified secondary metabolite compounds.** Secondary metabolite compounds identified with UHP liquid chromatography/mass spectrometry and processed using Compound Discoverer (Version 3.2). Compounds were then annotated with online and in-house accurate mass libraries (ChemSpider, etc.) and MSn mass spectral libraries (mzCloud, mzVault with in-house and public databases) and compared to authentic reference standards using the same instrument method to obtain accurate mass, MS2 fragmentation, and retention time matches.
(CSV)

## Acknowledgments

I would like to thank Elizabeth Leonard and Dr. Nishanth Therayil at the Clemson University College of Agriculture, Forestry, and Life Sciences Multi-user Analytical Lab. I would like to especially thank J. Forest Palmer, lab manager extraordinaire and trained Callery pear assassin, for his help with collecting leaf tissues for analysis.

## Author contributions

**Conceptualization:** Jessica Hartshorn.

**Investigation:** Jessica Hartshorn.

**Methodology:** Jessica Hartshorn.

**Project administration:** Jessica Hartshorn.

**Resources:** Jessica Hartshorn.

**Writing – original draft:** Jessica Hartshorn.

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
