## [Decision Letter · Decision Letter 0]

Dear Dr. Hartshorn,

Thank you for submitting your manuscript to PLOS ONE. After careful consideration, we feel that it has merit but does not fully meet PLOS ONE’s publication criteria as it currently stands. Therefore, we invite you to submit a revised version of the manuscript that addresses the points raised during the review process.

Your study shows some merit but one of the reviewers pointed out serious issues that I must agree with. The main problem is that the manuscript has no direct tests of allelopathy or tests from isolated chemicals in sensitive plants. This turns most of the discussion and conclusion speculative. The experiment is also limited to a few samples without replication in a greenhouse or different seasons/years. This makes the results very limited and the discussion and conclusion must be rewritten addressing all the limitations and being limited to the descriptive nature of the work. 

Thus, if all the points raised by reviewers were addressed and the discussion and conclusions were rewritten to address their speculative nature I can consider another reviewing round sending it to reviewers.

We look forward to receiving your revised manuscript.

Kind regards,

Fabricio Jose Pereira, Ph.D.

Academic Editor

PLOS ONE

Journal Requirements:

3. Please upload a copy of Supporting Information Table 1 & 2 which you refer to in your text on page 6.

Reviewers' comments:

Reviewer's Responses to Questions

**Comments to the Author**

1. Is the manuscript technically sound, and do the data support the conclusions?

Reviewer #1: Yes

Reviewer #2: No

2. Has the statistical analysis been performed appropriately and rigorously?

Reviewer #1: Yes

Reviewer #2: Yes

3. Have the authors made all data underlying the findings in their manuscript fully available?

Reviewer #1: Yes

Reviewer #2: No

4. Is the manuscript presented in an intelligible fashion and written in standard English?

Reviewer #1: Yes

Reviewer #2: Yes

Reviewer #1: Secondary metabolites of the invasive tree, Callery pear (Pyrus calleryana), provide support for the empty niche theory of invasion

General comments:

- In this work the Author compare the secondary metabolite profiles between Callery pear (Pyrus calleryana Decne.), black cherry (Prunus serotina) as the native comparison, and wild peach (Prunus persica) to serve as a non-native but non-invasive representative.

- Evidence suggests that the CP aggressive spread and negative impacts are due to allelopathic chemicals present throughout plant tissues which reduce herbivory and add unique allochthonous inputs to the soil, thereby reducing germination of native species and furthering Callery pear’s domination on the landscape.

- She found 32 unique compounds compared to our other two species tested, with 23 of these compounds belonging to flavones. Callery pear suggests downstream effects on native plants and arthropod communities, and provides evidence for the empty niche theory of invasion.

- These findings are novel and very important in terms of gaining knowledge about plant invasion mechanisms and from the conservation point of view

- I agree that the results only confirms the ‘empty niche theory’, but only partially. Of course there is no congener species in the US, suggesting existence of empty niche, but I think the other aspects of species biology are also important for invasion: the scleromorphic leaves (thick and glossy, related to drought resistance), quick growth (up to 90 cm per year), high generative and clonal growth potential – root suckering, and seed dispersal by animals. The unique ‘battery’ of secondary metabollites seems help the plant additionally get an advantage in resistance to abiotic and biotic stresses, after establishment.

Methods:

- please provide more information about study areas (geographical location, maps, maybe and environmental factors that can potentially affected the metabolite profiles

- statistical analyses – please provide the literarure and software used.

Detailed comments:

Line 94 I suggested to state the hypotheses research questions directly to link the comparison of secondary metabolities

Line 99 …”S.C”. – please use full geographic names and geographical location (s) of the study area for clarity

Reviewer #2: The study provides a comparative analysis of secondary metabolite profiles between invasive (Pyrus calleryana) and native/non-invasive plants (Prunus serotina and P. persica), offering preliminary insights into potential biochemical mechanisms underlying plant invasion. The focus on flavonoids and their ecological implications aligns with contemporary research on allelopathy and niche theory. However, the current data are insufficient to robustly support the conclusions and most of the discussion are speculative. My key concerns are as follows:

1. Lack of direct evidence: The study identifies metabolite differences but does not demonstrate their functional roles (e.g., allelopathy, microbial interactions).

2. Speculative discussions: Claims about allelopathic effects or soil microbial impacts are unsupported by experimental data.

3. Narrow scope: Sampling was limited to leaf tissue in one season, omitting root exudates, rhizosphere microbes, and temporal variation.

To substantiate the conclusions, the authors should conduct complementary experiments:

i) Microbial community analysis: Compare rhizosphere soil microbiomes between P. calleryana and native plants to test for flavonoid-driven shifts.

ii) Common garden bioassays: Apply flavonoids to native plants to quantify growth inhibition or microbial community changes.

iii) Competition trials: Co-culture P. calleryana with native species under controlled conditions to assess competitive outcomes linked to metabolite production.

iv) Expand metabolite profiling: Include root tissues and seasonal sampling to evaluate spatial/temporal dynamics of flavonoid production.

**Do you want your identity to be public for this peer review?** For information about this choice, including consent withdrawal, please see our Privacy Policy

Reviewer #1: No

Reviewer #2: No

---

## [Author Response · Author response to Decision Letter 1]

9 May 2025

Reviewer #1: Secondary metabolites of the invasive tree, Callery pear (Pyrus calleryana), provide support for the empty niche theory of invasion

General comments:

I agree that the results only confirms the ‘empty niche theory’, but only partially. Of course there is no congener species in the US, suggesting existence of empty niche, but I think the other aspects of species biology are also important for invasion: the scleromorphic leaves (thick and glossy, related to drought resistance), quick growth (up to 90 cm per year), high generative and clonal growth potential – root suckering, and seed dispersal by animals. The unique ‘battery’ of secondary metabollites seems help the plant additionally get an advantage in resistance to abiotic and biotic stresses, after establishment.

Thank you for your helpful comments. I did add L265-266 and 268-270 to the discussion to revisit some of the biological reasons that CP is so successful.

Methods:

- please provide more information about study areas (geographical location, maps, maybe and environmental factors that can potentially affected the metabolite profiles

Thank you for the suggestion. I added a map (Figure 1) with all sites represented. I also added a table with the number of each species found at each site, since they were not evenly distributed. I have also added Lines 101-110 with environmental and climate information about the sites. L234-236 in the discussion addresses the limitations of collecting in the same types of sites and call for expanded collections across different site types.

- statistical analyses – please provide the literarure and software used.

Thank you! This was a total oversight on my part. I added this information to L161 and also added Pang et al. 2022 as the citation.

Detailed comments:

Line 94 I suggested to state the hypotheses research questions directly to link the comparison of secondary metabolities

Added L94-96: “… and predicted that the profile of CP would be significantly different from that of both peach and black cherry.”

Line 99 …”S.C”. – please use full geographic names and geographical location (s) of the study area for clarity

Changed to South Carolina

Reviewer #2:

The study provides a comparative analysis of secondary metabolite profiles between invasive (Pyrus calleryana) and native/non-invasive plants (Prunus serotina and P. persica), offering preliminary insights into potential biochemical mechanisms underlying plant invasion. The focus on flavonoids and their ecological implications aligns with contemporary research on allelopathy and niche theory. However, the current data are insufficient to robustly support the conclusions and most of the discussion are speculative. My key concerns are as follows:

Lack of direct evidence: The study identifies metabolite differences but does not demonstrate their functional roles (e.g., allelopathy, microbial interactions).

Speculative discussions: Claims about allelopathic effects or soil microbial impacts are unsupported by experimental data.

Narrow scope: Sampling was limited to leaf tissue in one season, omitting root exudates, rhizosphere microbes, and temporal variation.

To substantiate the conclusions, the authors should conduct complementary experiments:

i) Microbial community analysis: Compare rhizosphere soil microbiomes between P. calleryana and native plants to test for flavonoid-driven shifts.

ii) Common garden bioassays: Apply flavonoids to native plants to quantify growth inhibition or microbial community changes.

iii) Competition trials: Co-culture P. calleryana with native species under controlled conditions to assess competitive outcomes linked to metabolite production.

iv) Expand metabolite profiling: Include root tissues and seasonal sampling to evaluate spatial/temporal dynamics of flavonoid production.

Thank you for the thoughtful comments. I fully agree that, without these additional experiments, the argument for allelopathy is tenuous. In my original discussion I tried to steer clear of specifically suggesting allelopathy in CP and give the study limitations which prevent me from making those claims. The empty niche theory of invasion does not equate directly to allelopathy, so I believe my original claims of evidence in favor of this theory still hold water. Unfortunately, I do not have the capacity to conduct these experiments as this was a side quest for a project which has since ended. However, there are currently projects underway including greenhouse controlled experiments to look at establishment and competition among CP and other species. In the updated discussion, I reference the studies outlined by R2 as necessary future research to confirm allelopathy in CP.

L234-236: In my original discussion I address the limitations of a single season of sampling and suggest that future research do additional sampling to look at seasonal changes to the SM profile.

Discussion additions:

L201-204: added next steps of quantification of SM

L237-241: added future research to apply flavonoids to natives and non-native non-invasive plants to look at both the effects on plant growth and defenses as well as the effects on rhizosphere communities

L252-255: added future research needed to look at SM of different tissue types

L266: added “and tissue types”

Updated Supp. Table info in paragraph to match style.

I found that all info is actually in Supp Table 1 so I removed the reference to Supp Table 2 and updated that to Supp Table 1. I also made sure to match the reference for the table to the guidelines.

Updated References heading to match style.

Updated Statistical analyses heading to match style.

Added “Supplemental information” section with appropriate table title and description. I made sure this matches the style guidelines as well.

Added L111-112 regarding permits for collections.

3. Please upload a copy of Supporting Information Table 1 & 2 which you refer to in your text on page 6.

See above comment re: supplemental tables.

---

## [Decision Letter · Decision Letter 1]

Secondary metabolites of the invasive tree, Callery pear (Pyrus calleryana), provide support for the empty niche theory of invasion

PONE-D-24-59000R1

Dear Dr. Hartshorn, 

We’re pleased to inform you that your manuscript has been judged scientifically suitable for publication and will be formally accepted for publication once it meets all outstanding technical requirements.

Kind regards,

Fabricio Jose Pereira, Ph.D.

Academic Editor

PLOS ONE

Additional Editor Comments (optional):

Reviewers' comments:

Reviewer's Responses to Questions

**Comments to the Author**

Reviewer #1: All comments have been addressed

2. Is the manuscript technically sound, and do the data support the conclusions?

Reviewer #1: Yes

3. Has the statistical analysis been performed appropriately and rigorously?

Reviewer #1: Yes

4. Have the authors made all data underlying the findings in their manuscript fully available?

Reviewer #1: Yes

5. Is the manuscript presented in an intelligible fashion and written in standard English?

Reviewer #1: Yes

Reviewer #1: in the current version of the manuscript, the author has addressed all my comments and has made appropriate corrections to the manuscript. At this stage, I have no further comments on the work

**Do you want your identity to be public for this peer review?** For information about this choice, including consent withdrawal, please see our Privacy Policy

Reviewer #1: No

---

## [Editor Report · Acceptance letter]

PONE-D-24-59000R1

PLOS ONE

Dear Dr. Hartshorn,

I'm pleased to inform you that your manuscript has been deemed suitable for publication in PLOS ONE. Congratulations! Your manuscript is now being handed over to our production team.

Kind regards,

on behalf of

Dr. Fabricio Jose Pereira

Academic Editor

PLOS ONE